

# A scalable and modular reservoir implementation for large scale integrated hydrologic simulations

Benjamin D. West[1], Reed M. Maxwell[2], Laura E. Condon[1]

[1]Department of Hydrology and Atmospherics Sciences, The University of Arizona, Tucson, AZ 85721, USA

[2]Department of Civil and Environmental Engineering, Princeton University, Princeton, NJ 08544 USA

*Correspondence to*: Ben West (benjaminwest@arizona.edu)

**Abstract.** Recent advancements in integrated hydrologic modeling have enabled increasingly high-fidelity models of the complete terrestrial hydrologic cycle. These advances are critical for our ability to understand and predict watershed dynamics especially in a changing climate.  However, many of the most physically rigorous models have been designed to

focus on natural processes and do not incorporate the effect of human built structures such as dams. By not accounting for these impacts, our models are limited both in their accuracy and in the scope of questions they are able to investigate. Here we present the first implementation of dams and reservoirs in ParFlow-an integrated hydrologic model. Through a series of idealized and real world test cases we demonstrate that our implementation (1) functions as intended, (2) maintains important qualities such as mass conservation, (3) works in a real domain and (4) is computationally efficient and can be scaled to large

domains with thousands of reservoirs.  Our results will improve the accuracy of current ParFlow models and enable us to ask new questions regarding conjunctive management of ground and surface water in systems with reservoirs.

## 1 Introduction

Reservoirs are critical infrastructure that support water supply for agricultural, municipal, and commercial use, provide flood

control benefits,  generate power, support recreation, and more (Graf, 1999; Ho et al., 2017; Wisser et al., 2013). There are an estimated 91,000 dams in the continental U.S. alone ("Infrastructure Report Card: Dams," 2017; Spinti et al., 2023). They drastically alter streamflow timing patterns, which impacts aquatic species, sediment transport, and water quality(Graf, 1999; Wisser et al., 2013)29/03/2024 19:45:00. In many parts of the world, reservoirs are also projected to be under increasing amounts of strain as they struggle to meet increasing human demand and environmental needs, all under uncertain

future conditions (Dettinger et al., 2015; Ho et al., 2017; Wisser et al., 2013; Zhou et al., 2018).

Given the widespread presence of reservoirs and their significant impact on stabilizing human water supplies and modifying river networks, it is imperative to have robust tools that can effectively represent their operations within hydrologic systems at large scales.  Decision support models are specifically designed to simulate networks of reservoirs and support decision-

making processes. Prominent examples include WEAP and RiverWare (Yates et al., 2005; Zagona et al., 2001). These models are tailored to address the complexities of interconnected reservoir systems and facilitate strategic planning and



management of water resources. There also exist highly specialized model implementations of important areas. For example, the Colorado Decision Support System (Fulp, 1995) represents all of the diversions and storage on the Colorado River system and is used for long term planning.


While decision support models have very sophisticated abilities to represent the complexities of human operations, they generally rely on simplified representations of the physical hydrology. For example, WEAP and Riverware represent reservoir networks as directional acyclic graphs, where each reservoir is associated with its own set of operating rules dictating the timing and volume of water releases. These models establish a flow pattern in which the outflow from one
reservoir serves as the inflow for another reservoir, often incorporating time delays or other mechanisms to account for factors like irrigation use or other outflows from the system. This approach is good at capturing human-driven operations and decision-making aspects of reservoir networks but may not adequately account for the dynamics and complexities of the natural environment especially under changing conditions.

On the other end of the spectrum there is increasing focus on representing human operations in hydrologic models (i.e. models that were primarily designed to understand hydrologic systems). MODFLOW, WaterGAP, PCR-GLOBW, and the National Water Model are all hydrologic models that now include reservoirs (Blodgett, 2019; Harbaugh, 2005; Kim et al., 2020; Müller Schmied et al., 2021; Sutanudjaja et al., 2018). These models handle the incorporation of reservoirs in a variety of ways. In the case of MODFLOW, it can be coupled with the decision support model MODSIM, which involves the
addition of reservoir nodes to the stream network. These nodes account for storage and release of water by exchanging information between MODFLOW and MODSIM at each timestep (Valerio, n.d.). PCR-GLOBW and WaterGAP, on the other hand, estimate water use from various sources, including irrigation and reservoirs. They account for the water consumed by these sources by removing it from the system. Return flows, if any, are then reintroduced into the appropriate locations (Müller Schmied et al., 2021; Sutanudjaja et al., 2018). Finally, the National Water Model designates locations as
lakes and reservoirs to track inflow. It utilizes parameterized functions such as weir or orifice functions to determine the release amounts from these reservoirs.

Although reservoir operations have been added into several physical hydrology models, it is worth noting that they are still absent from fully integrated hydrologic models that solve the surface and subsurface simultaneously and account for variably
saturated flow, like ParFlow and ATS (Coon et al., 2019; Kuffour et al., 2020; Maxwell et al., 2015; O'Neill et al., 2021). These models are very powerful because they solve physically based equations for water and energy fluxes for both the surface and subsurface at the same time. They are good at representing the vadose zone and can capture evolving systems. Integrated hydrologic models possess unique strengths in examining the intricate interactions between difference components of the environment, especially in changing conditions. For instance, ParFlow has been used investigate the
connections between groundwater flow and transpiration partitioning (Maxwell & Condon, 2016).

Previous work in models that do have reservoirs shows that they can have significant impacts on the hydrologic cycle beyond their direct streamflow alterations. For example, in a NOAH-HMS model of the Poyang Lake basin a 2.7% increase in total ET was observed, mostly attributed to direct ET from the reservoir surface (Wei et al., 2021). The same study also

observed a slightly higher groundwater table which resulted in slightly wetter soils and a 0.7% decrease to groundwater recharge due to the reduced hydraulic gradient at the surface. Globally, a WaterGAP model showed a reduction of discharges to oceans by 0.8% due to ET that occurred from reservoir surfaces. Low flow rates also decreased by 10% on a quarter of global land area due to reservoir releases redistributing flow temporally(Doell et al., 2009). Note that these studies exclude smaller reservoirs, which make up the majority of reservoirs but hold a minority of the overall storage. This exclusion may

mean some effect sizes are underestimated.

Adding reservoir operations to integrated physical hydrology models would have several benefits. Perhaps most importantly, it would increase their accuracy in managed systems and expand the range of research questions that can be addressed. Additionally, previous research has shown that reservoirs not only alter streamflow regimes but can have much more far-

reaching impacts on the watershed as a whole (Graf, 1999; Hwang et al., 2021; Wei et al., 2021; Wisser et al., 2013; Zhou et al., 2018). Integrated hydrologic models are the best tools to evaluate reservoir impacts on these complex integrations, as they directly solve the underlying physics of the complete system. Lastly, they may be able to provide new, generalizable understanding and characterization of reservoir operation either through controlled experiments or inference on existing data.

Here we present the first implementation of reservoirs for the integrated hydrologic model ParFlow. We provide detailed specifications on how reservoirs are represented within the ParFlow framework (Sections 2.1 – 2.3) and how a model user can interact with the reservoir module that is developed here (Section 2.4). Proof of concept for the new reservoir module is demonstrated with four sets of test cases run on idealized domains scaling from one reservoir up to 10,000 reservoirs (Sections 3.1 – 3.4). Our scaling tests illustrate performance up to the national scale demonstrating the feasibility of this

approach for integration into large scale simulations.

## 2 Reservoir Implementation Methods

In this section, we provide a detailed description of the implementation of reservoirs in ParFlow. We start by explaining the basic functioning of ParFlow in section 3.1. Then, we provide a high-level overview of the choices we made and the

rationale behind them in section 3.2. In section 3.3, we illustrate how the reservoirs interact with ParFlow's grid via an intake and a release cell. Finally, in section 3.4, we provide a comprehensive list of the fields needed to define a reservoir and demonstrate the user interface for incorporating reservoirs into a ParFlow model.



## 2.1 ParFlow

ParFlow is a gridded PDE-based hydrologic model that solves both the surface and subsurface flow systems simultaneously(Kollet & Maxwell, 2006; Kuffour et al., 2020; Maxwell et al., 2015). In the subsurface ParFlow solves the mixed form of Richards' Equation (Equation 1) (Celia et al., 1990; Richards, 2004) for variably saturated flow, with a general source/sink term added to account for transpiration, wells, and other fluxes.

$$S_s S_w(h) \frac{\partial h}{\partial t} + \varphi \frac{\partial S_w h}{\partial t} = \nabla \cdot \mathbf{q} + q_r \quad (1)$$

Here $h$[L] is the pressure head, $t$ is time, $S_s$ [L$^{-1}$] represents the specific storage, $\phi$ [–] denotes the porosity, $S_w$ [–] represents the relative saturation, $q_r$ [T$^{-1}$] represents a general source/sink term. The flux term $\mathbf{q}$ [LT$^{-1}$] is based on Darcy's Law:

$$\mathbf{q} = -K_s(x)\, k_r(h)\, [\nabla\,(h + z)\, \cos\,\theta_x + \sin\,\theta_x] \qquad (2)$$

Where $K_s$ (x) [LT$^{-1}$] represents the saturated hydraulic conductivity tensor, $k_r$ [–] represents the relative permeability, $z$ [L] denotes the elevation, and $\theta$ represents the local angle of topographic slope between the relevant grid cells, with $S_x$ and $S_y$ denoting the slopes in the x and y directions, respectively. The angles $\theta_x$ and $\theta_y$ are calculated as $\theta_x = \tan^{-1}(S_x)$ and $\theta_y = \tan^{-1}(S_y)$.

The van Genuchten (1980) relationships are employed for the relative saturation ($S_w(h)$) and permeability functions ($k_r(h)$)(van Genuchten, 1980).

There are multiple solvers available for overland flow in ParFlow, including both the kinematic and diffusive wave equations. Our implementation of reservoirs works with the kinematic wave equation, which is derived by applying continuity equations to pressure and flux and uses Manning's equations for the depth-averaged velocity $v_{sw}$ [LT$^{-1}$] (Kollet & Maxwell, 2006).

$$k \cdot (-K_s\,(x)k_r(h) \cdot \nabla\,(h + z)) = \frac{\partial\,\|h,0\|}{\partial t} - \frac{\nabla \cdot \|h,0\|}{v_{sw}} + \lambda q_r(x) \qquad (3)$$

With $k$ being the unit vector in the upwards ($z$) direction and $\lambda$ being a constant equal to the inverse of the vertical grid spacing. Note that the $\|h,0\|$ operator means select the larger value between $h$ and 0. This operator means that surface flow only occurs when pressure is greater than 0. The head used in the overland flow equation is the same as the head used in Equations 1 and 2 this free surface overland flow boundary condition allows for the integrated solution of the surface and subsurface.

Equations 1-3 are discretized onto a grid and solved in a non-linear, fully implicit fashion using a Newton-Krylov solver.
This integrated, implicitly solved system allows for correct resolution of interactions between surface and subsurface flow and mass conservation across the whole system.

More simply, because the surface and subsurface systems are solved in an integrated fashion. ParFlow also does not require the user to specify the location of a stream network a priori. Flow emerges when enough water has converged to result in
surface pressures above zero. This type of approach is also taken by other integrated hydrologic models such as Hydrogeosphere (Brunner & Simmons, 2012).

Unfortunately, as reservoirs rely on large human-built infrastructure that is not statically situated (think the operating of the release gate), they do not emerge naturally in the same way as streams. Instead, our implementation of reservoirs makes
extensive use of the source sink terms in Equation 3. Specifically, anthropogenic releases of water are added as source-sink flux terms to the top of the domain

## 2.2 Reservoir Module Overview

At the heart of our implementation is the addition of a reservoir module to ParFlow's code that enables the creation of
reservoir objects. ParFlow models are built and run using model definition files, which may be written in either Python or Tcl. Reservoir objects can be added individually as model attributes in any language ParFlow supports or en masse through a CSV file using ParFlow's Python SDK. Each reservoir object has a set of attributes (shown in Figure 1) such as location and minimum and maximum capacity, along with a state variable that tracks its storage throughout the simulation.

In addition to the reservoir object, the module includes several new subprocesses that run to account for the intake and release of water from reservoirs. These subprocesses modify the functioning of the two grid cells, the *Intake Cell* and *Release Cell*, at which the reservoir is located (henceforth all mentions of reservoir object fields will be denoted in *Uppercase Italics* to distinguish from cases we are simply discussing reservoirs broadly). At these locations the reservoir module overrides ParFlow's normal physics to ensure water only flows downstream when the reservoir's operating rule says
there should be a release. Rule curves are used to determine the amount of water that should be released from the reservoir.

As illustrate in Figure 1, at every timestep (1) water flows into the reservoir and the storage is updated, (2) the amount of water that is released from the reservoir is determined based on the reservoir's *Storage,* and (3) these releases are applied as fluxes at the release cell.






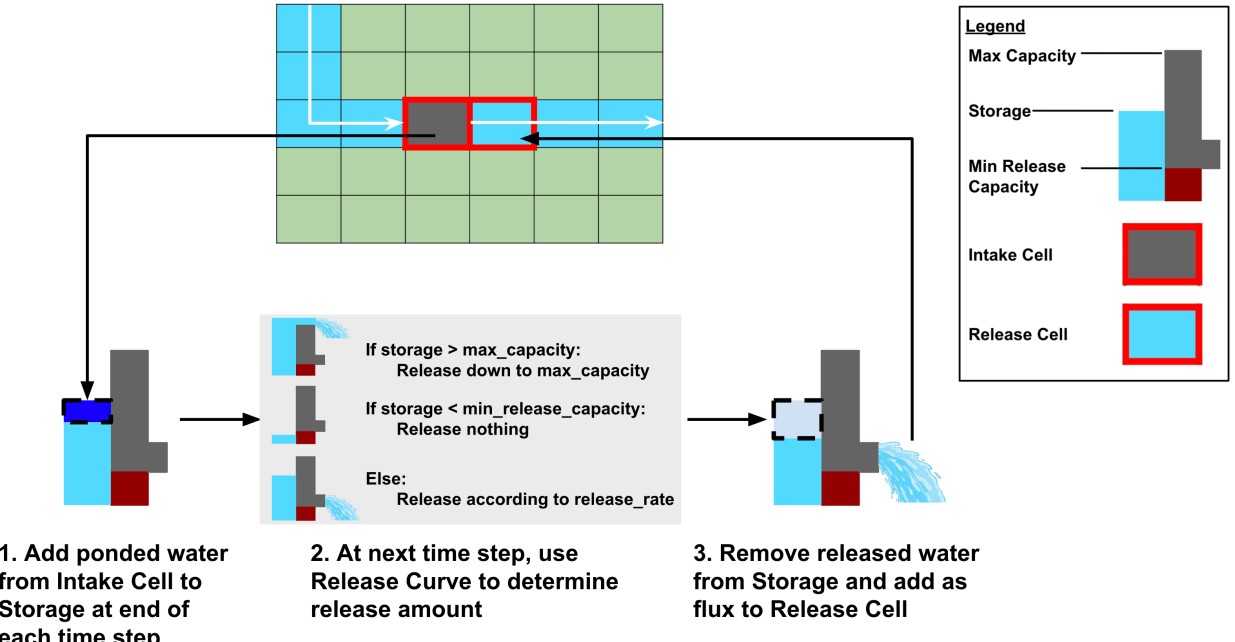

**Figure 1:** *Conceptual depiction of a reservoir ParFlow's grid. Here we see how water comes off an Intake Cell and into Storage, to be released later at the Release Cell in accordance with the reservoirs release curve.*


Our goal for our implementation was to choose a flexible representation that could represent a variety of use cases. ParFlow models are run at many scales spanning hillslope scale simulations up to continental models covering millions of square kilometers. Additionally, reservoirs are highly abundant and often do not have good data available about their operating rules. When data is available, there is not one consistent format or way of defining the operations that everyone uses. For these reasons, we prioritized simplicity and efficiency of implementation to establish a core set of functionalities that could easily be extended in the future as the reservoir data landscape shifts.

The representation we chose is also designed to be computationally efficient making it feasible to scale up to national scale simulations. By imposing fluxes directly at the cells, we avoid convergence issues associated with simulating the raising and lowering of the reservoir's stop gate. We also chose simple initial release curves, which enables us to develop a framework for adding more complex curves and test how reservoirs impact runtime performance.

### 2.3 Intake and Release cells

As shown in Figure 2, each reservoir has an *Intake Cell* and a *Release Cell*. The *Release Cell* is located directly downstream of the Intake Cell based on the topographic slopes. If a cell has multiple faces with outward slopes, the steepest gradient is selected to designate as the *Release Cell*.



The *Intake Cell* captures all incoming water and store it in the reservoir. At every timestep any ponded water is removed from the surface of *Intake Cells* and added to reservoir storage. To prevent any water from flowing directly through the cell before it can be scraped, the slopes of the outlet faces are set to zero during a preprocessing step. While this approach is effective for the *OverlandFlow* and *KinematicWave* surface flow solvers, it is not applicable to the *DiffusiveWave* solver, as water can diffuse out as it begins to pond. For this reason, our implementation of reservoirs does not support *DiffusiveWave.*

To capture incoming water, we have implemented a process that iterates through all reservoir *Intake Cells* at each timestep. This looping is achieved through ParFlow's concept of subgrids, which enables efficient parallelization. If a reservoir's *Intake Cell* has a pressure head $h$ greater than zero, $h$ is reset to zero and the corresponding volume of water (calculated by Equation 4) is added to the reservoir's storage:

$$\text{Volume of water to be added } = \text{ h} \cdot \text{dx} \cdot \text{dy} \quad (4)$$

Where $dx$ and $dy$ are the grid cell dimensions in the x and y directions.

In some cases, a single *Intake Cell* is not sufficient. This occurs when channels are two cells wide, which can happen even for narrow channels under particular geometries. In these cases, a *Secondary Intake Cell* can be designated. *Secondary Intake Cells* function the same as other *Intake Cells*, with the water in them being diverted to reservoir *Storage*. However, *Secondary Intake Cells* do not have associated *Release Cells*.

*Release Cells* also have special operations. At each timestep, the reservoir module iterates over all the reservoirs and calculates how much water each should release based on the current storage and the reservoir release rules as follows: If a reservoir's storage is below its *Min Release Capacity*, no water is released. If its *Storage* exceeds its maximum capacity, all excess water is released. Otherwise, the release rate is determined by the reservoir's *Release Rate* attribute. The release flux is then added as a flux at each reservoir's *Release Cell,* and at the end of the timestep, the release flux is removed from the reservoir's *Storage*. Note that as ParFlow's solver is implicit the size of release does not necessarily impose a constraint on the timestep size.

## 2.4 Reservoir fields and user interface

As mentioned in section 2.3, each reservoir is represented by an object that has multiple attributes and state variables. In order to add a reservoir to a model, the user must either define these fields, or they must be inferable from the other fields (as in the case of the location of the release cell). These fields may be seen in Table 1.



**Table 1:** Reservoir fields and their descriptions

| Field Name | Short description |
| --- | --- |
| Name | A unique name for the reservoir |
| Intake_X | The X location of the reservoir's intake cell in ParFlow grid coordinates |
| Intake_Y | The Y location of the reservoir's intake cell in ParFlow grid coordinates |
| Secondary_Intake_X | The X location of the reservoir's secondary intake cell in ParFlow grid coordinates, if it has one |
| Secondary_Intake_Y | The Y location of the reservoir's secondary intake cell in ParFlow grid coordinates, if it has one |
| Release_X | The X location of the reservoir's release cell in ParFlow grid coordinates. May be calculated during an automated preprocessing step. |
| Release_Y | The Y location of the reservoir's release cell in ParFlow grid coordinates. May be calculated during an automated preprocessing step. |
| Storage | The amount of water in the reservoir. The value supplied by the user determines the initial storage of the reservoir, but this field gets updated as the simulation runs. In units of $L^3$. |
| Min_Release_Capacity | The capacity below which the reservoir will stop releasing water in units of $L^3$ |
| Max_Capacity | The capacity above which the reservoir will release all excess water in units of $L^3$ |
| Release_Rate | The rate at which the reservoir releases water in units of $L^3T^{-1}$ |
| Release_Amount_In_Solver | An internal variable that tracks how much water to take out of the reservoir at each timestep. Does not need to be specified by the user. |

To add reservoirs with these fields to a model, users can take one of two approaches. The first approach is to use the new *ReservoirPropertiesBuilder* class (henceforth referred to as the Builder) that we added to ParFlow's Python software development kit. This is expected to be the most common option for domains with more than one reservoir. To do this, users only need to provide the Builder with the location of a file containing their reservoir data, with one reservoir per line. The Builder will then add each row as a reservoir to the model. For an example snippet refer to the appendix, Figure A1.

Using the *ReservoirPropertiesBuilder* class to add reservoirs to a ParFlow model comes with several advantages besides concision. Namely, the class performs automatic preprocessing to calculate the locations of release cells, edit the slope files at intake cell locations appropriately, and check for errors such as the existence of multiple reservoirs on one cell. While



these tools are available to users with non-Python model definitions such as Tcl, they require a separate Python script to be called before the model is run, whereas using the *ReservoirPropertiesBuilder* class handles these steps automatically.


The second approach to add reservoirs to a ParFlow model is to add each reservoir attribute one by one as model attributes in the ParFlow run script (either Tcl or Python). This approach is less concise, and each reservoir will require 8+ lines of code. However, it may be preferable for domains with a small number of reservoirs, and it allows users to add reservoirs to older workflows, such as models written in Tcl. An example snippet may be seen in the appendix, Figure A2.


Our goal with this user interface is to require as little overhead and manual intervention as possible. To this end, we have selected fields that align with existing reservoir databases such as GRanD and ResOpsUS(Lehner, Liermann, et al., 2011; Lehner, ReidyLiermann, et al., 2011; Steyaert et al., 2022) (Maxwell et al., 2015; Steyaert et al., 2022). Furthermore, we plan to enable users to inform the Builder that they are using these formats, allowing it to read them directly and convert

them automatically in addition to the other preprocessing tasks we are handling.

**3 Test Cases**

We implemented three types of test cases to demonstrate the performance of the new reservoir module. In Section 3.1, we present test cases conducted on an idealized domain, where we verify the proper functioning of reservoirs in a simplified

setting with known expected outcomes. In Section 3.2, we run an ensemble of conditions on this domain to verify reservoirs respond as expected to perturbations. In Section 3.3 we add a reservoir to a real domain and show some analysis tools available for it. Lastly, in Section 3.4, we examine performance test cases to assess the computational impact of incorporating reservoirs into a domain.

**3.1 Idealized Test Cases**

Idealized test cases validate the functionality of various aspects of our reservoirs. Specifically, we aim to demonstrate that reservoirs release the appropriate amount of water according to their defined release curves, intake the correct amount of water, and maintain mass conservation within the domain. Furthermore, this case provides a qualitative illustration of how reservoirs respond to variations in domain parameters and how they impact streamflow.


For the first test case we simulate an idealized domain in three conditions (1) *draining*, (2) *filling*, and (3) *periodic rainfall*. These cases represent common scenarios encountered by reservoirs and are designed to capture behavior across the entire release curve. Tests were conducted on a simplified tilted-v domain, with a single reservoir placed midway down the channel, as illustrated in Figure 2.  The domain is 9.77  square km, roughly one-tenth the size of a typical HUC12 watershed

(*National Hydrography Dataset | U.S. Geological Survey*, n.d.).





The reservoir *Max Capacity* is 7 MCM (million cubic meters), and *Min Release Capacity* is 5 MCM. The *Max Capacity* equates to 28.24 inches of rainfall across the domain in a year, which is roughly equal to the annual rainfall in Minnesota (Kunkel, 2022).


**Figure 2: Idealized illustration of the tilted-v test case adapted from (Farley & Condon, 2023). The domain is 25x25x2 grid cells, each 125x125x5 meters in size. This leads to each hill slope being 1500 meters from domain edge to channel. The slope along the channel is 0.01 while the slopes perpendicular to the channel are 0.1. The sides and bottom are no flow boundaries.**

The domain is discretized into 25 columns, 25 rows, and 2 layers, resulting in a total of 1,250 grid cells (each grid cell is 125 by 125 m). The domain is homogenous with a saturated hydraulic conductivity (K) value 0.000001 m/h, (a reasonable K value for clay). The slope along the channel is 0.01, and the slopes perpendicular to the channel are 0.1. This configuration effectively promotes the immediate runoff of water into and then down channel,

Zero flux boundaries are applied along the bottom and sides of the domain. Rainfall is applied along the top boundary and varied based on the test case. The reservoir in the domain was placed at the middle cell of the domain, which lies in the channel. It was given a *Max Capacity* of 7 MCM and a *Min Release Capacity* of 5 MCM.

Table 2 describes the initial *Storage*, *Release Rate* and rain rate for each of the three test cases as well as the expected result.


**Table 2:** Idealized test case configurations

| Test Case | Initial | Release | Rain Rate | Expected Behavior |
|---|---|---|---|---|



| | Storage (MCM) | Rate (MCM/h) | (m/h) | |
|---|---|---|---|---|
| **Draining** | *7.03* | *0.01* | *0* | An immediate release of water down to *7 MCM*, followed by the reservoir continuing to release water at its *Release Rate* until it reaches its *Min Release Capacity*. |
| **Filling** | *3.5* | *0.01* | *0.005* | The reservoir should gradually fill up until it reaches its *Max Capacity*, after which it will start releasing water at the same rate it receives rainfall. |
| **Periodic Rainfall** | *6* | *0.01* | Alternating between *0.005* and *0* every hour | We should observe fluctuations in the reservoir's storage in response to the periodic rainfall, with an overall gradual upward trend. |

Figure 3 shows the reservoir storage, intake rate, and release rate over time for all three test cases. Overall, the results align with the expected behavior described in Table 2. In the filling case (Figure 3a), the storage of the reservoir, increases until it reaches its *Max Capacity*. At this point, all incoming water is released, resulting in the release rate (dotted orange line) becoming equal to the intake rate (solid orange line). In the draining case (Figure 3b), there is an initial surge in the release rate as excess water is discharged, which can be seen in the large early value reached by the dotted orange line. This is followed by a steady decline in storage (blue line) at the specified *Release Rate* until the *Min Release Capacity* is reached.

Once the *Min Release Capacity* is attained, the reservoir ceases to release water. Lastly, in the periodic rainfall case in panel C, we observe fluctuations in reservoir storage (blue line) with upticks corresponding to when the intake rate (solid orange line) is positive and downticks corresponding to when there is no intake, with a gradual overall upward trend. We also note that in the draining case (Figure 6b), the amount of time it takes for the reservoir to empty is completely consistent with the reservoir's *Release Rate*, indicating that the internal storage being correctly accounted.






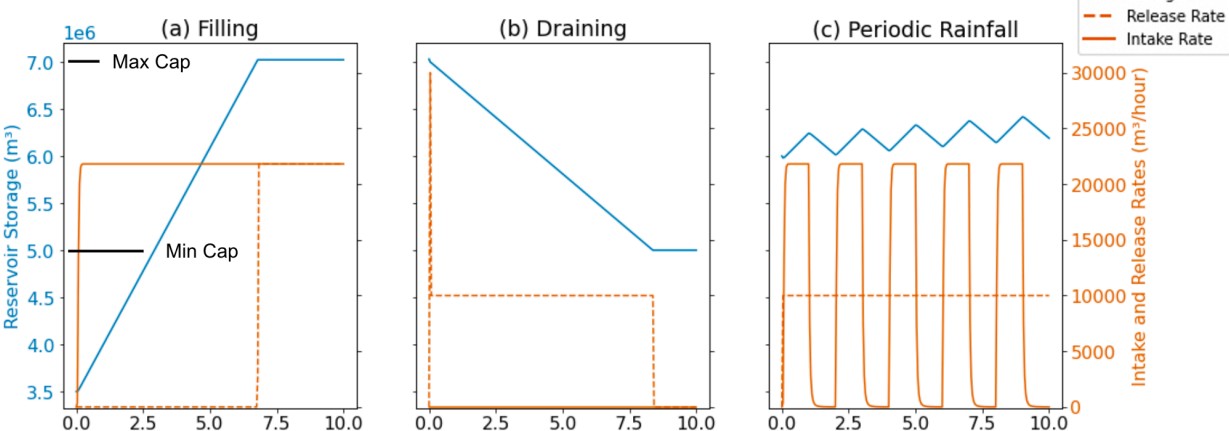

**Figure 3** *Idealized Test Case Results, for the Filling (a), Draining(b), and Periodic Rainfall (c) cases.  For each case reservoirs storage (blue line), intake rate (solid red line), and release rate (dashed orange line) are plotted as a function of time for the 10-hour simulation.*
*The reservoir's max capacity (Max Cap) and min release capacity (Min Cap) are shown on the y axis of subplot a.*

Overall Figure 3 validates that the reservoirs are functioning as intended, adhering to the expected behaviors outlined in Section 2.2 and Table 2. These findings provide confidence in the accurate representation of the release curves across their entirety.


**Section 3.2 Ensemble Case**

Next, we conducted an ensemble of simulations on the idealized domain to demonstrate (1) the capability of reservoirs to handle a wide range of scenarios, and (2) validate their expected behavior in response to perturbations.  The ensemble involved varying the *Release Rates* of the reservoir and the rain rates within the domain. A range of *Release Rates* from 0 to
0.01 MCM/hour and rain rates from 0 (no rainfall) to 0.01 m/hour (a heavy storm) were selected. For each combination of *Release Rate* and rain rate, the simulations were run for a duration of 1 day at an hourly timestep. We initialized the reservoir with a *Max Capacity* of 5.5 MCM for all simulations to ensure consistency across the ensemble.

Figure 4 shows the relationship reservoir storage at the end of the simulation and release and rain rate. We anticipated a
linear relationship between storage and both release and rain rates, leading to a plane-like surface in the 3-D plot. As expected, the plotted points exhibit this linear pattern, with higher rain rates corresponding to increased storage and lower *Release Rates* associated with higher storage values. The simplicity of both our domain and our release curves are what drive the strong linearity observed in these relationships. Given more complex topography, rainfall patterns and operating curves this plot would be less linear.






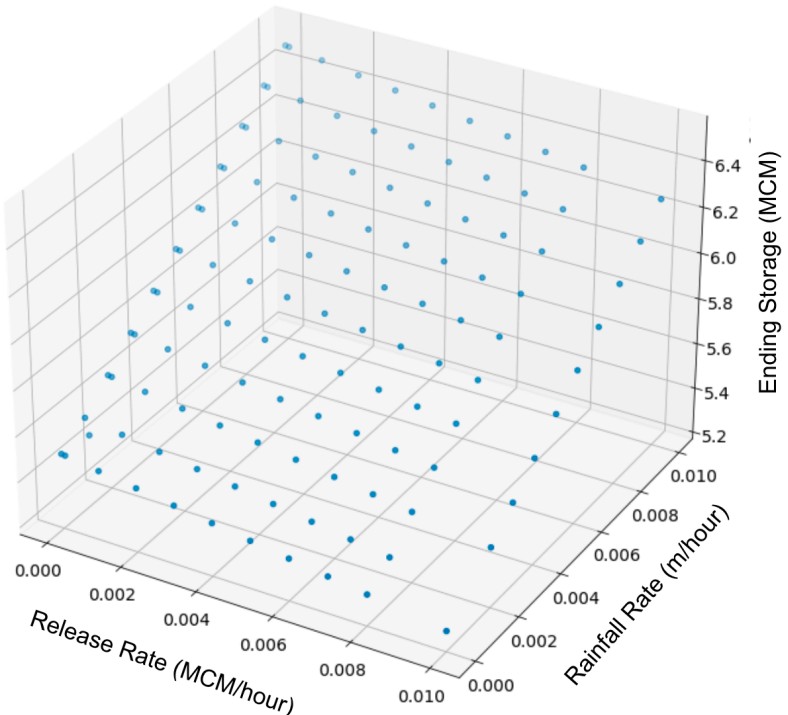

**Figure 4:** *Storage change after 1 day vs. rainfall and release rates. Each point represents one member of the simulation ensemble. Points are shaded to help give a sense of depth.*

To explore the relationship between rainfall and storage more quantitatively we compare the rate of storage change in the reservoir to the maximum possible inflow. Maximum possible storage change is calculated as the total land area upstream of the reservoir multiplied by the rain rate. In practice we don't expect the reservoir storage changes to be exactly equal to this maximum possible storage change because (1) not all water drains to the reservoir, (2) there can be time lags between precipitation and reservoir inflow, (3) some of the precipitation will infiltrate into the subsurface. However, in this simple

domain we expect these values to be close.

Figure 5 plots the relationship between simulated storage change and maximum possible storage change. As expected, we see a linear relationship between storage change and maximum possible storage change that falls that slightly below the 1-to-1 line.



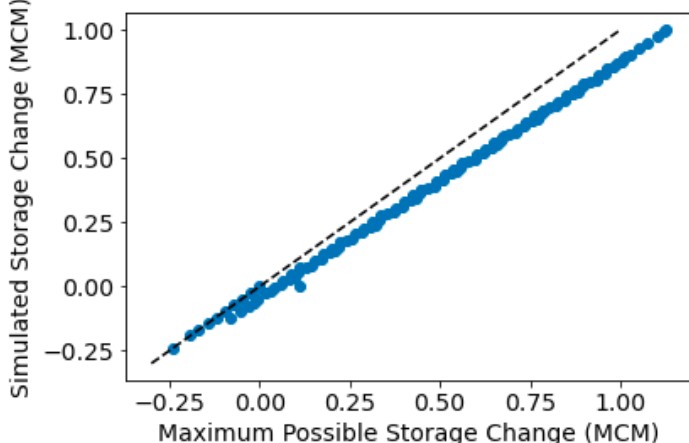


***Figure 5:*** *A comparison of actual storage change after one day to the maximum possible daily storage change. The 1-1 line is shown in dashed black for reference.*

This consistency with a simple mass balance indicates that under a wide variety of conditions reservoirs both intake and
release the appropriate amount of water.  This is further evidence that reservoirs are effective under a wide range of conditions and exhibit the desired behaviour in response to variations in release and rain rates.

### 3.3 Real world test case

Next, we conduct a test on a real domain, whose full description can be seen in (Leonarduzzi et al., 2022). This test is
intended to demonstrate that the implementation is robust enough to work in existing workflows and not just idealized cases. It also shows that even very simple operating rules modify streamflow signals in the ways we know reservoirs do, and demonstrates the ability to analyze water storage in a domain holistically.

The real domain we chose is the East-Taylor watershed. This 767 square-mile watershed lies in the headwaters of the
Colorado river, near the continental divide. It contains two rivers, the East and the Taylor, which converge right before they exit the domain which can be seen in Figure 6. The Taylor River is regulated by the Taylor Park Dam about halfway along its reach, while the East River is unregulated. This clear split between the regulated and unregulated tributaries makes for a natural comparison. The East-Taylor ParFlow domain was previously developed by xx et al. For full documentation of the domain the reader is refereed to insert citation.


For our test cases we add the Taylor Park Dam to the existing domain. We simulate two different modifications to the baseline run of the domain in 2003, which only simulates natural flow. In the first we add a reservoir where the Taylor Park Dam is that releases no water at all. In the second we have this same reservoir release water at the average inflow rate for



2003. In both cases the reservoir is sized such that reaching either the minimum release storage or storage capacity were not
concerns, as the intent was to observe the effect of regulation on the domain and ensure the mass-accounting held up once more features, such as meteorological forcing, were turned on.

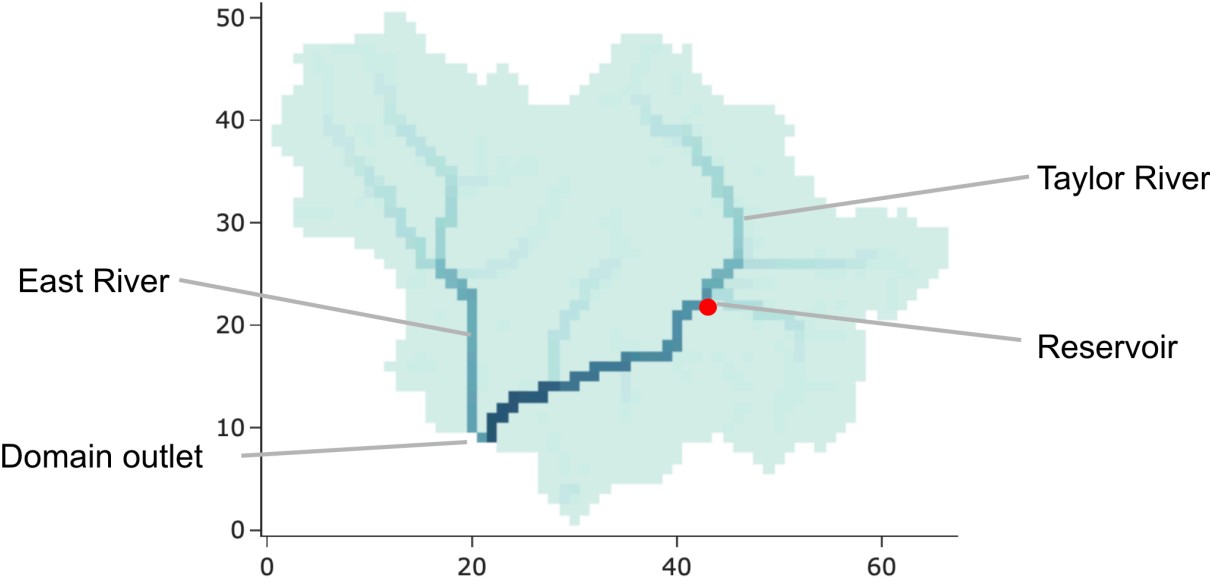

**Figure 6** A map of the East-Taylor Watershed with the locations of the East River, the Taylor River, the reservoir, and the domain outlet noted


The results of the simulations can be seen in Fig 7. In the case that no water is released we can clearly see the reservoir blocking flow immediately downstream on the surface, with a gradual increase as you head downstream as runoff accumulates from the drainage area downstream of the reservoir. The hydrograph at the reservoir outlet shows this as well, with an almost flat signal close to zero. Note that there can still be some flow in the reservoir outlet flow in this case that is
contributed by cells downstream of the reservoir. Moving to the Taylor River domain outlet we see a dampened signal compared to natural flow, with proportionally lower base and peak flows. Lastly, we see no change in the East River outlet flow, just as we expected.

In the case that we release water at a constant rate the story is a bit different. At our selected timestep the flow downstream
of the reservoir is higher than naturalized conditions. This is because a time of low flow is being supplemented by a release of water. At the reservoir outlet we again see a mostly flat signal, but at the release rate and not at zero. Moving down to the Taylor river outlet we now see a modulated flow signal, with higher baseflows but lower peaks. This type of modulation is exactly the effect reservoirs tend to have in reality. Lastly, we again see no change to the flow of the East River.


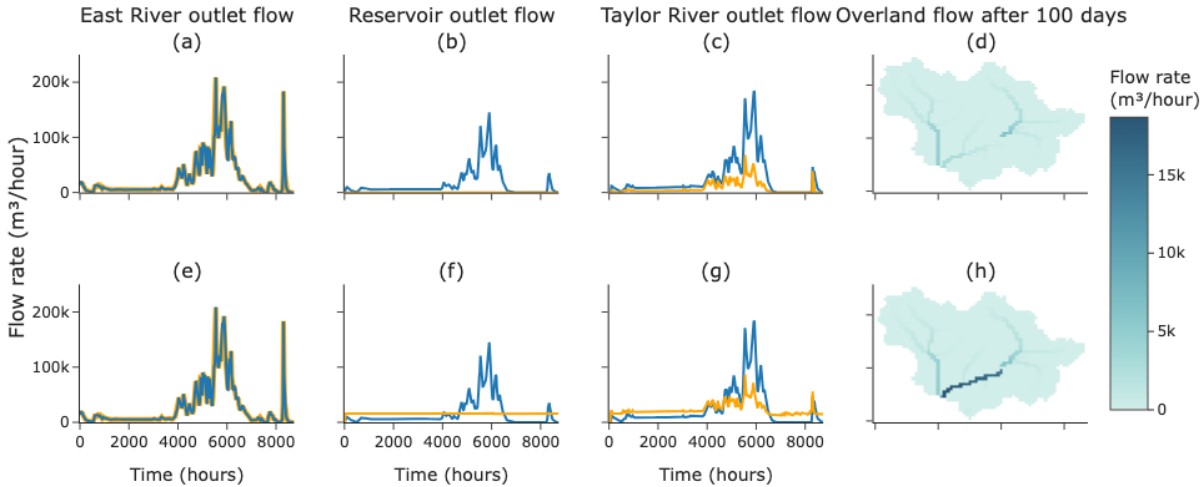

**Figure 7** *Real world test case results for both simulations. The top row (a,b,c,d) pertains to the case of adding a reservoir that never releases, while the bottom row (e,f,g,h) pertains to the case of a reservoir that releases at a constant rate. Column 4 (d,h) depicts the overland flow on the surface 2400 hours in to the simulations, with lighter, yellower shades representing more flow. Columns 1, 2, and 3 are hydrographs of the flow at different locations on the surface, with the blue lines representing the base case of no reservoir and the orange lines representing the simulation results. Column 1 (a,e) is the flow at the East River's outlet, column 2 (b,f) is the flow at the reservoir's outlet, and column 4 (c,g) is the Taylor River's outlet.*


We also demonstrate an example storage analysis throughout the year on the domain for the simulation of a reservoir that releases at a constant rate, which can be seen in Figure 8. Our module allows us to monitor changes in subsurface storage as well as reservoir storage concurrently. Interestingly, for this domain the change in reservoir storage and subsurface storage throughout the year are of roughly the same magnitude.






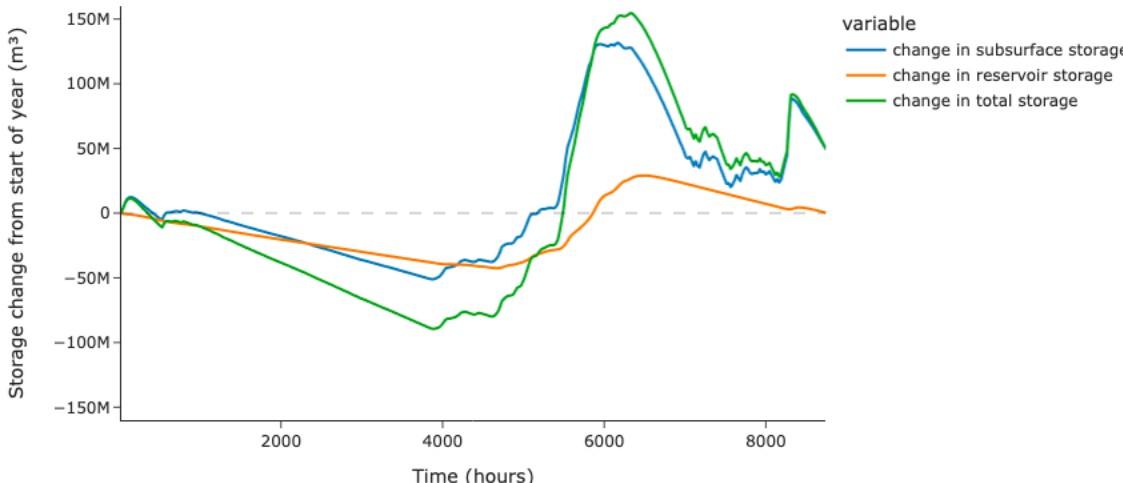

**Figure 8** *A storage analysis for the simulation of a reservoir releasing at a constant rate. The blue line tracks the change in subsurface storage from the start of the year, the orange line tracks the change in reservoir storage from the start of the year, and the green line tracks the change in sum of both of those storages.*


These test cases demonstrate proof of concept for our implementation in a real domain. Also, this shows that even adding a reservoir with the simplest possible operating impacts streamflow similarly to real reservoirs. Finally, we show that adding reservoirs in a domain allows for holistic analysis of water storage.


### 3.4 Performance Scaling Tests

Finally, we conduct a series of scaling tests to evaluate the computational cost associated with incorporating many reservoirs into a domain. These are crucial for understanding the potential impact on compute time, as the number and complexity of experiments that can be conducted are often constrained by computational limitations.


The performance tests were completed on a sloping slab domain (shown in Figure 9) consisting of 500 by 500 grid cells in the x and y direction (625,000 lateral grid cells total) and two layers. The lateral resolution of the domain is 10m and the cell thickness is 5m. As with the first test domain, no flow boundaries were applied on the bottom and sides of the domain and a uniform saturated hydraulic conductivity value of 0.000001 m/h (typical of clay) was used for the entire subsurface. As with

the previous test case we are creating a relatively impermeable subsurface to promote runoff as we are really focusing on the surface system here. The same three physical cases from the first test case are used—draining, filling, and periodic rainfall—to investigate whether certain physical scenarios would result in significantly longer compute times.



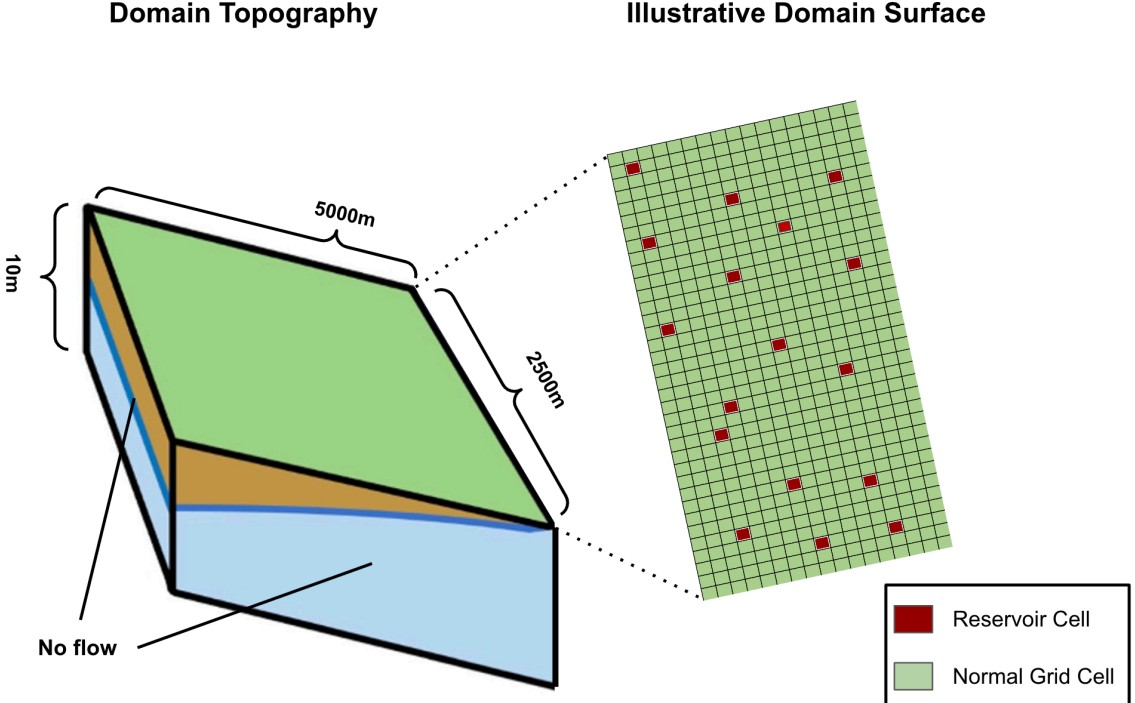

**Domain Topography**        **Illustrative Domain Surface**

**Figure 9:** *Illustrative sloping slab domain with no reservoirs on it adapted from (Farley & Condon, 2023). The left side (domain topography) shows the shape of the domain, while the right side (domain surface) depicts how reservoirs are scattered at random on the surface of the domain. The number of grid cells depicted is just illustrative for the sake of legibility. The actual domain size is 500x500x1 cells, with the cells having dimensions 10x10x5 meters. No flow boundaries are set along the sides and bottom.*

For each case, we conducted 8 simulations with an increasing number of reservoirs randomly placed throughout the domain, ranging from 0 to 10,000. We chose 10,000 reservoirs as it represents a number larger than any of the currently available datasets of historical releases for the US. Each simulation was run for a total of 10 hours of simulation time. A range of simulation time components are tracked including total simulation time, pre-processing time, time spent on the first timestep, time spent on the second timestep, and time spent on the subsequent timesteps.


Figure 10 displays performance test results. The top row (Figure 10a-c) depicts the ending surface pressures for each case for the scenario with 10,000 reservoirs added to the domain. Here green indicates surface water, brown indicates a water table below the surface, white indicates a pressure near zero, and red indicates a reservoir's *Intake Cell*. In the filling case (Figures 10a), reservoirs are capturing most of the rainfall that falls on the domain and there are white streaks downstream of the

reservoirs as they block flow. For the draining case (Figure 10b), the domain is mostly saturated, except for the areas that are



upstream of any reservoirs. In the draining case there is no rainfall on the domain and all the water comes from the reservoir releases (Refer to Table 2). Finally, the periodic rainfall case (Figure 10c), has the greatest spatial heterogeneity. Again, this is to be expected as reservoirs will be filling and draining at different rates depending on their upstream areas and the presence of reservoirs upstream of them. In general, the streamflow increases from left to right across the domain as both
reservoir releases and rainfall are aggregated moving downstream.

Figure 10d-f show the computational costs of each of the test cases for an increasing number of reservoirs. Overall, we show that the additional computational burden is not extreme for any of the cases. This is apparent by noting the trend in the top line height from left to right in panels D, E, and F, which increases ~30% for filling, ~100% for draining, and ~15% for
periodic rainfall when 10,000 reservoirs have been added to the domain. In terms of the scaling, we note that the x-axis is a log scale. This means that the mostly linear looking increases are logarithmic, and the sharp jump in the *Draining* case at 10,000 reservoirs represents an approximately linear increase.

There are some indications that costs in real simulations are likely to be smaller than this. The most extreme increase of
~100% is only seen when adding 10,000 reservoirs to the draining case. This results in a quick transition from an area that was drying up to one that was saturated and had some surface flow. This is an expensive physical situation to solve, and only happens because of the extremely high density of reservoirs on the domain that would never be seen in real models. Also, on all three domains, we see that the increase in cost is mainly occurring in the later timesteps. This indicates that the overhead to run the reservoir module is minimal, and the cost comes from increasing the physical complexity. Model run
times are usually constrained by the most complex process to simulate, and as this domain is extremely simple without them, reservoirs are far more likely to be that process. In real domains it is far more likely that other complex physics are occurring, and reservoirs are not the limiting factor.

It's also informative to look at the computational demand of the first- and second-time steps (shows in green and orange) as
these are steps where we expect large changes as reservoirs are being activated. For instance, one might expect that the simultaneous activation of multiple reservoirs at the same time step could lead to slow convergence due to the resulting pattern of fluxes being applied. However, our results indicate that this is not the case and there are no strong trends between the computational demand for the first- and second-time step and the number of reservoirs applied. This is promising for future simulations of more realistic releases that vary in time.





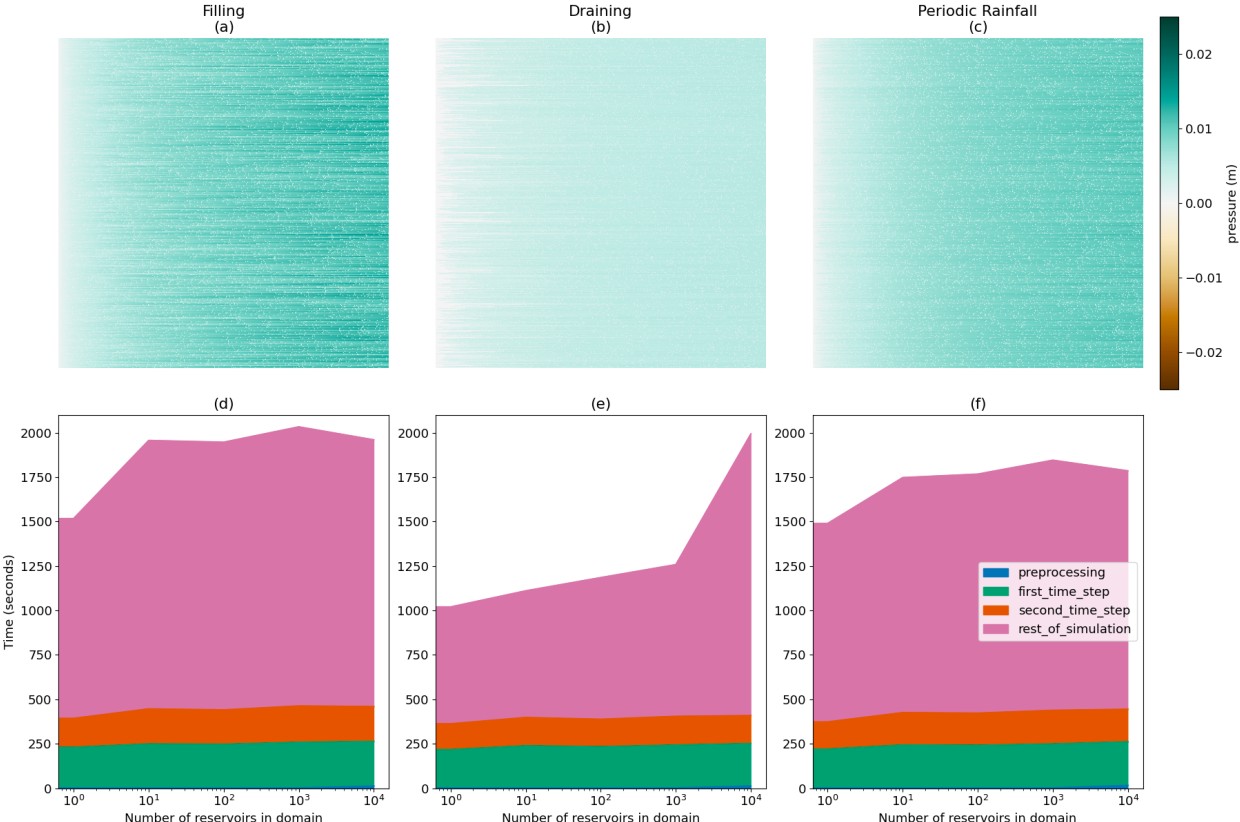

**Figure 10:** *The results of our performance scaling tests. Left column (panels a and d) correspond to the filling case, center column (panels b and e) corresponds to the draining case, left column (panels c and f) corresponds to the periodic rainfall case. Top row (panels a, b, and c) depicts the surface pressure at the end of the simulation when 10000 reservoirs were placed on the domain. Water flows left to right. Bottom row (panels d, e, and f) depicts how simulation performance scales as reservoirs are added to the domain, with each component of the simulation shown in a different color.*

Overall, these findings demonstrate that the reservoir module itself incurs minimal additional computational cost for the simulation, even for large numbers of reservoirs. Additionally, none of the components show non-linear performance degradation as we add reservoirs to the domain, except for the pre-processing in the largest case which is likely due to machine limitations that could be alleviated by running on more cores. Furthermore, the overall degradation in performance time remains and is acceptable given that it is a one-time cost. This lack of degradation of the other components is promising, as small performance changes can dominate behaviour as problem sizes scale, causing even initially very cheap operations to cause significant overall slowdowns.



Still, it should also be noted that reservoir operations can create a hydrologic conditions that take longer to solve. As illustrated by the draining test case. Very large releases create sudden fluxes into the domain which can be harder for the model to solve. Thus, we conclude here that the reservoir module scales well and does not add significant computational cost at run time, which is crucial for supporting our goal of CONUS scale simulation. However, depending on reservoir

configuration reservoir releases can change computational demands significantly.

**5 Conclusions**

This paper presents a novel reservoir module for the integrated hydrologic model ParFlow. This addresses a previously significant gap in the platform. Our implementation takes a straightforward approach by overriding the physical behavior at

two grid cells per reservoir. This enables the storage and timed release of water based on a reservoir's release curve and is in line with other physical hydrology models such as MODFLOW, WaterGAP, PCR-GLOBW, and the National Water Model (Blodgett, 2019; Harbaugh, 2005; Müller Schmied et al., 2021; Sutanudjaja et al., 2018). We have also developed a user-friendly interface that simplifies the addition of reservoirs to a domain using existing data files.

The module presented here is fully integrated into the ParFlow code base and is designed to support future expansion. Potential future features include release curves that are time or storage-dependent (i.e. parameters that vary monthly), release curves that call user supplied models (e.g. a trained machine learning model), and automatic data loaders that can convert popular datasets like GRanD and ResOpsUS (Lehner, ReidyLiermann, et al., 2011; Steyaert et al., 2022) into inputs to the reservoir module.


Test cases demonstrate that our implementation (1) functions as intended, (2) maintains important qualities such as mass conservation, (3) works in a real domain and (4) is computationally efficient and can be scaled to large domains with thousands of reservoirs.

The new reservoir implementation provides two key advances. First, this will improve the ability of ParFlow models at any scale to match streamflows in managed systems. This will improve simulation performance especially for large domains and will allow new studies that consider management operations directly. Second, by implementing reservoirs into a fully integrated groundwater surface water model we can facilitated new approaches to explore water management from a more holistic perspective that is not possible with other models that already include reservoirs but simplify subsurface processes.

For example, quantifying the role of vadose zone exchanges on the hydrologic impacts shown with previous models (e.g. groundwater and soil moisture changes) (Doell et al., 2009; Wei et al., 2021), improving and generating reservoir rule curves via model inference, testing the impacts of hypothetical operation changes on the environment, and estimating the change falling groundwater levels could have on reservoir recharge in the future. Many of these questions require some combination





of high physical realism and either large scale simulations or large simulation ensembles, the approach presented here is

uniquely positioned to support these types of analysis.

## 5 Appendix

```
reservoirs_csv = "./reservoirs.csv"
reservoirs_builder = ReservoirPropertiesBuilder(model).load_csv_file(reservoirs_csv)
reservoirs_builder.apply()
```

**Figure A1:** *A Python code snippet demonstrating how to add reservoirs from a file to a ParFlow model. In this case the file is a csv, but*

*the builder supports any of the file formats supported by ParFlow's TableToProperties class.*

```
model.Reservoirs.Names = "reservoir_1"

model.Reservoirs.reservoir_1.Intake_X = 1560
model.Reservoirs.reservoir_1.Intake_Y = 1560
model.Reservoirs.reservoir_1.Release_X = 1690
model.Reservoirs.reservoir_1.Release_Y = 1560
model.Reservoirs.reservoir_1.Max_Capacity = 7000000
model.Reservoirs.reservoir_1.Current_Storage = 5500000
model.Reservoirs.reservoir_1.Min_Release_Capacity = 4000000
model.Reservoirs.reservoir_1.Release_Rate = 10.
```
**Figure**

**A2:** *A Python code snippet demonstrating how to add reservoirs to a ParFlow model without loading them from a file*

## 6 Code Availability

ParFlow's code, which is where the reservoir module was added, is available at https://doi.org/10.5281/zenodo.4816884.
Code for the idealized and performance test cases can be found at https://github.com/westb2/reservoir_test_cases. Code for
the real-world test case can be found in the original paper describing the domain https://doi.org/10.5194/hess-2022-345

## 7 Author Contributions

Author contributions are as follows: Conceptualization; RMM, BW, LEC; Formal Analysis; BW, LEC, RMM; Funding
Acquisition; LEC, RMM; Investigation; BW, LEC, RMM; Methodology; BW, RMM, LEC; Supervision; LEC, RMM;
Visualization; BW, LEC; Validation; BW, LEC; Writing-Original Draft; BW, LEC; Writing-Review and Editing; BW, LEC.

## 6 Acknowledgements

The authors would like to acknowledge that AI was used in the writing process. Specifically, for early drafts Chat-GPT was
asked to edit paragraphs for clarity and grammar. All of these were subsequently edited by us, and in later drafts all editing
was done by the authors themselves.



**7 Competing Interests**

The authors have no competing interests to declare.

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
