# Peer review of "A scalable and modular reservoir implementation for large scale integrated hydrologic simulations"

_EGUsphere, 2024_

## Referee Comment (RC1)

[referee-annotated manuscript omitted]

---

## Author Comment (AC1)

This manuscript is well written and presents a solid summary of work to incorporate reservoirs as simple idealized model objects. I have a few comments that may be worth addressing, but they are all suggestions and should not hold up publication of this work.

I've attached a copy of the pdf with inline comments.

*Thank you for taking the time to provide thoughtful feedback! We appreciate your positive evaluation and have included responses to each of your specific suggestions below in italics.*

Line 81: Not sure this sweeping statement is justified
*We agree and will modify this line to be less general and more specific to the particular advantages of Integrated hydrologic models*

Lines 178-181: I feel like the design philosophy of starting with simplicity and extensibility in mind could be stated more succinctly at the start of this section
*We agree and will revise this per your suggestion.*

Lines 189-190:  scraped?
*We are using "scraped" here to denote the process of removing water from the domain and placing it into storage. We agree this is not the most intuitive word choice and will instead use the "diverted into reservoir storage" in the revised version.*

Lines 241-242: Nice touch – might want to mention this up in the intro? Accessibility is king and it would be good to make clear that this approach aligns with existing data.
*Thank you! We worked really hard to make this accessible and agree that this could be highlighted a bit more.   In response to this comment and two similar comments, we will add a short paragraph describing our design philosophy, along with a brief description of our strategy of allowing our reservoirs to have their actual released amount calculated by a separate process in order to support both existing and future ways of representing operations.*

Line 425, Figure 9: Those are some really little reservoirs if these are 10m grid cells! Is this maybe too much of a toy example?
*We agree that 10m is small for a reservoir, however we would like to note that the purpose of this test case is only for testing scaling.   Our goal here is to measure the efficiency of the underlying additions to Parflow's code both in terms of extra calculations being done, and extra memory being stored and accessed. Neither of these should be tied to the dimensions of individual grid cells.  In this case the size of each grid cell does not impact the difficulty of the problem, it is really the number of grid cells being solved and the number of reservoirs we are applying, therefore, changing the size of our grid cells does not actually change the size of the problem.  That said, we do agree that it may appear odd to some readers to have such small grid cells. We will add a few sentences to the description of this test case noting that we are scaling the number of unknowns in the problem and that similar results would be expected regardless of the grid cell sizes.*

---

## Author Response (AR1)

**Editor response:**

**Reviewers are split, with one recommending "accept as is" and the other requesting a major revision to address validation of the reservoir scheme, which may have arisen due to some misinterpretation on the study's primary goal. Please revise the manuscript to help the guide the reader toward the correct interpretation of the work, perhaps with expanded commentary on how results should be interpreted, and additional discussion on how one might later parameterize the proposed scheme to capture realistic reservoir operations. Comments relating to improved "accuracy" (e.g., last line of the abstract) perhaps set up an expectation for demonstration of improved accuracy that the work has yet to deliver.**

Thank you for handling our manuscript and for addressing the differences between the two reviewers. In the revised manuscript we have responded to all the specific comments of the reviewers and have also expanded our discussion throughout to provide more context and help improve the interpretation of our work. We have provided our point-by-point response below but would like to highlight the larger changes we mad in response to your comments here.

With respect to the suggestion to help guide the reader to the correct interpretation of the work we made several revisions. Firstly, we added a paragraph to our introduction more explicitly describing the goal of this implementation as well as going into more detail about what we intend to demonstrate with our test cases. Secondly, when introducing each of our test cases, we have added a sentence or two more explicitly trying to state in somewhat plain language what we are trying to demonstrate. Thirdly, we have significantly revised our conclusions (please refer to lines 225-229, 332-333, 373, 434-435, and 517-529 of the revised manuscript to see these changes).

We have also added discussion on how one might later parameterize the proposed scheme to capture realistic reservoir operations in our discussion of the reservoir release curve implementation. We added the following paragraph describing future types of release curves this format will support.

> *"Even this simple release curve will allow for some useful simulations. However, the limited number of parameters used here does limit the complexity of operations that can be simulated. The module we have implemented here can easily be expanded support a wide variety of release curves that may be needed for different applications. It would also be possible in the future to develop tools to read in release time series directly and bypass the rule curve approach entirely. This approach would allow for direct simulation of datasets of historical releases.*

Finally, we revised the sentence in the abstract that mentioned 'improved accuracy' as we realize this was a bit vague. Also, through the edits to the introduction and conclusions that were previously noted we tried to improve the clarity around the purpose and potential for this module.

As a final note, due to an unfortunate interaction with our citation manager track changes shows a large deletion of almost the entire intro. We have fixed it to the extent that is possible. You can still see all the changes we made and simply have to ignore that large deletion.

Thank you for the time, and please let us know if you need any additional information.

**Reviewer 1 comments**

*Relevant revisions have been added in italics below the initial response to reviewer for every comment.*

**Line 81: Not sure this sweeping statement is justified**

*We agree and have modified this line to be less general and more specific to the particular advantages of Integrated hydrologic models*

*Relevant revisions: line 85 "could be useful"*

**Lines 158-159: I feel like the design philosophy of starting with simplicity and extensibility in mind could be stated more succinctly at the start of this section**
*We agree and have revised per you suggestions.*

*Relevant revisions: lines 263-264: "The use of a separate module allows us to start with a simple approach while building a framework that can easily be extended to more complicated formulations in the future"*

**Lines 189-190: scraped?**

*We used "scraped" here to denote the process of removing water from the domain and placing it into storage. We agree this is not the most intuitive word choice and have revised.*

*Relevant revisions: Lines 220-221: "removed from the domain and put into reservoir storage"*

**Lines 241-242: Nice touch – might want to mention this up in the intro? Accessibility is king and it would be good to make clear that this approach aligns with existing data.**
*Thank you! We worked really hard to make this accessible and agree that this could be highlighted a bit more. In response to this comment and two similar comments, we will add a short paragraph describing our design philosophy, along with a brief description of our strategy of allowing our reservoirs to have their actual released amount calculated by a separate process in order to support both existing and future ways of representing operations.*

*Relevant revisions: lines 90-95 and lines 225-229*
> *"Our initial implementation is quite simple, but it provides the scaffolding for future development.  For example, in our reservoir parameterizations we have intentionally selected fields that align with existing reservoir databases such as GRanD and ResOpsUS to facilitate future real-world simulations with these databases (Lehner, Liermann, et al., 2011; Steyaert et al., 2022) (Maxwell et al., 2015; Steyaert et al., 2022). We focus on idealized test cases in this manuscript but demonstrate scaling out to large number of reservoirs to demonstrate the feasibility of large-scale applications."*

> *"Even this simple release curve will allow for some useful simulations, such as the effect of limiting the release rate of a reservoir during a drought. However, one of the first extensions we have planned is to support more varieties of release curves to support a broader set of simulations. Now that we have functionality in place that does variable releases, it will be simple to make these either functions of time or more sophisticated functions of storage. This will allow for direct simulation of datasets of historical releases as well as simulations of expected future reservoir behavior directly in-situ with the environment. Currently such simulations must correct measures such as streamflow post-hoc, which requires sacrifices in accuracy."*

**Line 425, Figure 9: Those are some really little reservoirs if these are 10m grid cells! Is this maybe too much of a toy example?**

*We agree that 10m is small for a reservoir, however we would like to note that the purpose of this test case is only for testing scaling. Our goal here is to measure the efficiency of the underlying additions to Parflow's code both in terms of extra calculations being done, and extra memory being stored and accessed. Neither of these should be tied to the dimensions of individual grid cells. In this case the size of each grid cell does not impact the difficulty of the problem, it is really the number of grid cells being solved and the number of reservoirs we are applying, therefore, changing the size of our grid cells does not actually change the size of the problem. That said, we do agree that it may appear odd to some readers to have such small grid cells. We have a few sentences to the description of this test case noting that we are scaling the number of unknowns in the problem and that similar results would be expected regardless of the grid cell sizes.*

*Relevant revisions Lines 438-440: "Note that the fine spatial resolution (10m) used here is just to simplify our experimental setup, our results would be the same if the grid sizing were larger.*

Reviewer 2 comments

**This study implements a reservoir scheme into ParFlow, an integrated hydrological model that simulates many other hydrological components but lacks reservoirs and dams. Specifically, the study uses idealized and real-world situations to demonstrate the proposed scheme functions as expected, ensure mass conservation, and is computationally reasonable. The manuscript presents formulations on the new scheme and details how it works in idealized and real-world settings. The East-Taylor watershed is used as a test bed toward the end of the manuscript. Overall, I was excited about the topic and agreed to review, also because the work comes from two leading groups in hydrological modeling. After reading the manuscript, I didn't find my expectations to be met, primarily because the manuscript entirely lacks validation. Please see my detailed comments below.**

*We thank the reviewer for their careful review of our work. We are sorry to hear that the manuscript did not meet the reviewer's expectations. Based on the comments below we understand that there is a disconnect here between our intended purpose of the paper and the reviewer's impressions. We have provided a detailed response below which we hope will clarify our purpose and the reason we did not provide the type of validation that the reviewer was expecting. We would like to emphasize that we do provide validation and sensitivity analysis of the modeling capability we present here. We do not provide site specific parameter evaluation as that is a test of local model calibration and not the model capability itself.*

**A major concern I have on this manuscript is the lack of validation. I waited until the end to see if some validation (real validation with observed flow or reservoir storage variation) is presented, but I found none. Validation is fundamental to any modeling study, which applies to reservoir modeling as well. A number of previous studies I note below have presented validation across the US and worldwide with available data on reservoir storage and release. If this manuscript were to be published in HESS, the authors should demonstrate model applicability across a range of basins in the US (and even beyond) and present a detailed validation of whether the model reproduces observed flows in the downstream of dams and storage variations. Substantial data exist**

**for many reservoirs, at least those in the US (please see numerous studies I note below). Without such explicit validation, I'm not sure what the value of the present study is.**

*While we acknowledge the reviewer's concern regarding the validation of modeling studies, it's important to clarify that the primary objective of our study was not to develop watershed models, but rather to develop a generic reservoir modeling capability within ParFlow. As such the validation we provide in our paper is to validate the new modeling capability which we are presenting here. The goal of our test cases is to demonstrate that our implementation is sufficiently robust for integration into a broad range of practical applications. We would argue that the suite of test cases we provided do provide explicit validation of the new module's capability to faithfully implement reservoir releases that are specified by the user.*

*What the reviewer is suggesting here would not be a test of whether the model itself is functioning properly, but rather a test of whether the operating parameters that we input to the model are realistic. This is of course an important test in any real-world simulation but is not a test of the model itself rather a test of the parameterization and model calibration. As the reviewer points out there have been other studies focused on rule curve development and validation, this is really a separate issue.*

*We hope that the reviewer will appreciate this distinction and the fact that we did design our test cases and model validation very thoughtfully. Building generic model capabilities such as this is a large undertaking and before we can validate site specific parameters the model itself must be validated and rigorously tested. That is the goal of our paper.*

**Page 2: Literature review is very limited in scope and doesn't include many of the early studies; reservoir modeling has been there for over two decades and numerous studies have attempted to incorporate reservoir in different kinds of models; I suggest that the authors acknowledge the previous students and expand the introduction; for example: (Dang et al., 2020; Haddeland et al., 2006; Hanasaki et al., 2006; Hanazaki et al., 2022; Shin et al., 2019; Vanderkelen et al., 2022; Wada et al., 2014) and many more. Even Hanasaki et al. (2006), one of the very early studies, has not been mentioned.**

*We appreciate the suggestion and acknowledge that some early citations were overlooked. We have expanded our literature review in the revised manuscript to include these citations as well as mentioning the vastness of previous work we are building on.*

*Relevant revisions Lines 29-35:*

> "There is a large body of literature on reservoir operations and long history of reservoir simulation (Blodgett, 2019; Dang et al., 2020; Haddeland et al., 2006; Hanasaki et al., 2006, 2022; Harbaugh, 2005; Kim et al., 2020; Müller Schmied et al., 2021; Shin et al., 2019; Sutanudjaja et al., 2018; Vanderkelen et al., 2022; Wada et al., 2014). Today, decision support models that are specifically designed to simulate networks of reservoirs and support decision-making processes are some of the most commonly used tools. Prominent examples include WEAP and RiverWare (Yates et al., 2005; Zagona et al., 2001). These models are tailored to address the complexities of interconnected reservoir systems and facilitate strategic planning and management of water resources."

**Lines 158-159: "Many other models, such as VIC, CaMa-Flood, LEAF-Hydro, and mizuRoute all contain reservoir representation (Vanderkelen et al., 2022) ."**

Introduction: It is generally well-written but it overly emphasizes implementation of dams in ParFlow. I suggest that the authors place more emphasis on: why this is needed (given existing models) and what is the novel contribution of the study?

*Thank you for the suggestion we did not intend to overly emphasize ParFlow.  We have done our best to make this clearer in the intro and conclusion.*

*Relevant revisions: Throughout both intro and conclusion, all have already been referenced in other replies.*

**Page 5, Line 162 and elsewhere: how is dead, active, and flood storage considered in the model? Please describe in the manuscript.**

*We capture both dead and active storage using the minimum_release_storage attribute. Our model does not distinguish between flood storage and normal storage. We have clarified this in the revised manuscript.*

Relevant revisions: *see lines 221-223 "In this representation any storage level below Min Release Capacity is dead storage, and any storage above it is active storage. We do not explicitly represent flood storage. However this is a feature that could be added in future implementations."*

**Equation 4: Are reservoirs considered to be of rectangular shapes?**

*Reservoirs modify flow within a single grid cell (or two adjacent cells), effectively represented as rectangles. We have added text to this section noting that reservoirs can only be resolved to the grid resolution and that we are not explicitly representing bathymetry here.*

Relevant revisions: *see lines 198-200 "It should be noted here that the intake cell is not intended to represent the inundated area of the reservoir or its bathymetry. It is most downstream location in the reservoir where we link the reservoir storage object to the grid. In this approach we do not explicitly resolve the reservoir inundated area.  "*

**Line 257: Where are the "release curves" taken from? It is critical to elaborate this point.**

*The release curves used here are idealized and not drawn from observations. As noted in our* response above o*ur intent with the paper is to demonstrate a generic modeling capability. The purpose of the rule curve in this case is to demonstrate and test the model's capability to handle releases under a range of conditions.  We understand that this may not have been made clear in the initial manuscript and have revised the text accordingly here to note the source and purpose of the rule curves used here. Please see the following comment for where this text was revised.*

**Page 10, Line 267: How are the volumes of 7 MCM and 5 MCM determined? Please provide the basis for these numbers.**

*As noted above this is an idealized test case designed to test the capabilities of the reservoir module and not to match observations. We designed a reservoir sized to accommodate approximately a year's worth of rainfall (as mentioned in lines 266-267), with a minimum release storage chosen arbitrarily. The release rates were selected to reflect the expected intake rate of the reservoir during simulations. As previously noted, these decisions were deliberate as our tests were not aimed at replicating a real-world scenario, but rather at validating fundamental properties such as the reservoir's mass balance. Therefore, we believe that opting for rounded numbers within a physically realistic range allows readers to verify our expected outcomes using straightforward calculations. We have added additional clarification to the revised manuscript to make the purpose and design of our test cases clearer.*

*Relevant revisions: lines 282-288*

> *"We add a single reservoir to the domain with a Max Capacity is 7 MCM (million cubic meters), and Min Release Capacity is 5 MCM. The reservoir is sized to hold roughly one year of precipitation in the domain. The Max Capacity equates to 28.24 inches of rainfall across the domain in a year, which is roughly equal to the annual rainfall in Minnesota (Kunkel, 2022). The minim release capacity was set at 5MCM. This is intentionally set relatively close to the Max Capacity to ensure that our test runs could easily achieve reservoir storage below the minimum capacity threshold. For both the Max and Min Capacity the goal of our reservoir sizing is choose reasonable values where our test cases can easily demonstrate correct behaviour across the full range of operating scenarios. "*

**Line 275: there are certain model parameters noted here, which also seem arbitrary; the authors should present a sensitivity analysis to demonstrate that these parameters are reasonable/robust.**

*Here too we would like to emphasize the purpose of our testing which is exclusively to validate the performance of the reservoir module. The parameters in question do not directly interface with reservoirs therefore a sensitivity analysis of these parameters would not help our validation efforts. We want to emphasize here that we did conduct a sensitivity analysis of parameters directly linked to reservoirs and provide details of this in the following section.*

*Relevant revisions Lines 330-332: "Next, we conducted an ensemble of simulations on the idealized domain to demonstrate (1) the capability of reservoirs to handle a wide range of scenarios, and (2) validate their expected behaviours in response to perturbations, ensuring we haven't designed something that only works in a narrow parameter range. "*

**L331, "maximum possible storage": Is this calculated? Maximum storage is available in reservoir database such as the GRaND data.**

*The maximum possible storage is a user-supplied field. If the user opts to utilize GRaND data, it will align accordingly; however, if they choose a different dataset, it will align with that dataset instead.*

**Figures: Figures including 4 and 6 do not provide important information; these could be placed in a supplemental document.**

*The purpose of Figure 4 is to present a sensitivity analysis of parameters related to the reservoir, as requested in a previous comment. We believe this analysis is warranted; however, we understand that the purpose of this sensitivity analysis may not have been clear and have provided additional explanation of the purpose of these figures in the revised manuscript.*

*Figure 6 represents our real-life domain and provides important context for interpreting Figures 7 and 8.*

*Relevant Revisions:*

> *Lines 331-332: "(2) validate their expected behaviors in response to perturbations, ensuring we haven't designed something that only works in a narrow parameter range."*

*Line 437: "we used to conduct a sensitivity analysis of variables related to reservoir function."*

**Figure 5, "maximum possible storage": I was expecting some validation with observed data but seems like this is just a comparison of simulated vs. "calculated" max storage. Please present actual validation.**

*Please refer to our earlier response regarding the purpose of this test case.*

**Line 355-359: seems like this is AI-written and not checked by the authors; for example, it reads: "insert citation".**

*This line was not generated by AI; it was an oversight by one of the authors while gathering additional citations for the final edit. We have corrected this in the revised manuscript.*

*Relevant revisions lines 527-528: "(Lehner, Liermann, et al., 2011; Lehner, ReidyLiermann, et al., 2011; Steyaert et al., 2022). Furthermore, in accordance with our commitment to providing a smooth user experience,"*

**L359: xx et al.?**

*We will correct this in the revised manuscript.*

*Relevant revisions lines 527-528: "(Lehner, Liermann, et al., 2011; Lehner, ReidyLiermann, et al., 2011; Steyaert et al., 2022). Furthermore, in accordance with our commitment to providing a smooth user experience,"*

**Figure 7: please present validation with observations; rich observations are available from USGS, US Bureau of Reclamation, and US Army Corps of Engineers for US reservoirs.**

*As noted in our previous responses, validating against observations does not meet the purpose of our manuscript. We are not testing reservoir parameterizations (i.e. what would be needed to calibrate to observations) we are presenting a generic modeling capability and as such have provided validation and sensitivity analysis of that capability directly.*

*Our objective is to show that our implementation is feature-complete and ready for application in real domains. We will emphasize this in the revised manuscript.*

*Relevant revisions have already been referenced*

**Figure 9: I wasn't able to understand the point of this figure; looks like a supplemental figure.**

*This figure describes the domain for our performance test case. We will explicitly mention this in the figure caption to provide clarity.*

*Relevant revisions line 448" "Figure 9: Illustrative sloping slab domain used for our performance test, adapted from (Farley & Condon, 2023)."*

**L492: How is it novel? Please clearly describe this in the introduction and here.**

*Our implementation is the first support for representation of reservoirs in ParFlow's class of models (fully-integrated physical hydrology models). It will both allow people using this class of models to make more accurate models, and for people interested in reservoir modeling to ask new questions that these types of models are best suited to answer. We have further emphasized this in the intro and conclusion*

*Relevant revisions have already been referenced.*

**L498: how is it "user friendly", especially compared to existing reservoir models.**

*User friendly is of course a matter of opinion but we appreciate the comment and can be more explicit on this point. We have tried to better clarify what is user friendly about our module in the throughout the manuscript and we also revised the sentence in question as follows:*

*Lines 243-244 "This builder is one of main ways we support an easy user experience.."*

**Overall, this manuscript can be a valuable contribution only if sufficient validation is presented to demonstrate that the reservoir module can reproduce observed release and storage.**

*We respectfully disagree with this opinion. Model development is a very important (and time consuming and challenging) part of the scientific process. We feel it is vitally important the model capabilities are rigorously validated and documented before application. It is important that we value rigorous model development as this is what allows us to put faith in their later applications. Absolutely any future application of this tool will require site specific validation and evaluation of reservoir parameters. However, the validation the reviewer is suggesting is fundamentally different from what we presented here and is a test not of the model itself but of the site-specific model parameters.*

**Minor issues:**

**Line 23: there is a time stamp, perhaps put by AI**

*This has been fixed.*

**L243: please fix citations**

*This has been fixed*

**Line 265: n.d.?**

*This has been fixed.*